# Danish GPs’ Experiences When Managing Patients Presenting to General Practice with Symptoms of Acute Lower Respiratory Tract Infections: A Qualitative Study

**DOI:** 10.3390/antibiotics10060661

**Published:** 2021-06-01

**Authors:** Lise Bisgaard, Camilla Aakjær Andersen, Morten Sig Ager Jensen, Lars Bjerrum, Malene Plejdrup Hansen

**Affiliations:** 1Center for General Practice, Aalborg University, 9220 Aalborg, Denmark; caakjaer@dcm.aau.dk (C.A.A.); mjensen@dcm.aau.dk (M.S.A.J.); mph@dcm.aau.dk (M.P.H.); 2Section of General Practice and Research Unit for General Practice, Department of Public Health, University of Copenhagen, 1014 Copenhagen, Denmark; lbjerrum@sund.ku.dk

**Keywords:** general practice, acute lower respiratory tract infection, antibiotics, qualitative study

## Abstract

One of the most common indications for antibiotic prescribing in general practice is acute lower respiratory tract infections (LRTI). This study aimed to explore general practitioners’ (GPs’) considerations and experiences when managing patients with symptoms of an acute LRTI. Individual semi-structured interviews were conducted with seven GPs in the North Denmark Region from January to March 2020. Data were analysed by means of systematic text condensation. The analysis revealed four themes: (1) practicalities of assessing patients with LRTI, (2) assessment of the patient, (3) treatment decisions, and (4) patient expectations. The GPs described having developed individual diagnostic strategies and routines when managing patients with symptoms of an acute LRTI. However, a general assessment of the patient was essential to all the GPs and the diagnosis was seldom based on a single symptom or finding. Most GPs described having great faith in abnormal lung auscultation. The use of C-reactive protein testing served several purposes, such as deciding on the severity of the infection, prescribing antibiotics or not, and as a communicative tool. Diagnostic uncertainty is a driver of antibiotic use and clinical practice might benefit from the development of clinical prediction rules for diagnosing pneumonia.

## 1. Introduction

Antimicrobial resistance is a worldwide threat to public health. Problems with resistant bacteria escalate when antibiotics are used inappropriately and abundantly, and immediate action must be taken to reduce the overuse of antibiotics [1]. In Denmark, about 75% of all antibiotic prescriptions are issued in general practice [2]. Hence, great responsibility for the reduction of antibiotic use lies within this sector.

One of the most common indications for antibiotic prescribing in general practice is acute lower respiratory tract infections (LRTI) [3]. In 2018, treatment of pneumonia accounted for 11% of all antibiotic prescriptions in Danish general practice only exceeded by treatment of urinary tract infections (27%) and the unspecified indication “infection” (20%) [4]. The majority of acute LRTIs in general practice are due to acute bronchitis, which is considered a self-limiting infection. On the other hand, pneumonia is a potentially life-threatening infection that should be treated with antibiotics [5]. However, it can be difficult to differentiate pneumonia from other acute LRTIs resulting in overuse of antibiotics [6]. Additionally, a previous study from UK general practice found that practices prescribing fewer antibiotics for self-limiting acute respiratory tract infections had a slightly increased rate of pneumonia (and peritonsillar abscess) than practices with a higher antibiotic consumption [7].

In general practice, pneumonia is considered a clinical diagnosis [8], and clinical features like cough, fever, auscultation abnormalities, dyspnoea and chest pain are positive predictors of labelling patients with pneumonia [9,10]. However, no strict diagnostic criteria exist and diagnosing pneumonia clinically can be challenging [5,8]. Chest radiography can be ordered in general practice when suspecting pneumonia. Still, the relatively low sensitivity in detecting pulmonary infiltrates, especially in the early course of the disease, needs to be kept in mind [11].

In order to improve the validity of the diagnostic process point-of-care tests are often used in general practice. A point-of-care test is a test performed during or very close to the time of the consultation [12]. Evidence exists that point-of-care C-reactive protein (CRP) testing can help rule out pneumonia [13] and guidelines recommend CRP testing for patients with symptoms of acute LRTI [5]. However, a recent Danish study found that even a slightly elevated CRP (>11 mg/L) was positively associated with patients being diagnosed with pneumonia and consequently treated with antibiotics [14]. Several studies indicate overuse of antibiotics in general practice [6,15], but very few studies have scrutinised GPs’ reflections regarding the diagnosis and treatment of patients with symptoms of acute LRTIs.

Before developing new interventions, aiming at rational use of antibiotics in general practice, solid knowledge is needed about GPs’ practice and interpretation of the information available when assessing patients with acute LRTIs (e.g., patient history, symptoms, clinical findings, test results, and examinations). Therefore, this study aimed to explore Danish GPs’ considerations and experiences when managing patients presenting to general practice with symptoms of acute LRTI.

## 2. Results

A total of seven GPs were interviewed for this study. Each interview lasted between 38 to 120 min. Information about the participating GPs is provided in Table 1.

The GPs described how they often encountered patients with acute LRTI in their daily work. They described individual diagnostic strategies and routines—all with a pragmatic approach. The GPs sometimes strayed from their usual routine or interpretations as each patient presented different challenges demanding an individual approach. A general assessment of the patient was essential for all the GPs. The analysis revealed four analytical themes describing the different dimensions included in the general assessment: (1) Practicalities of assessing patients with LRTI, (2) assessment of the patient, (3) treatment decisions, and (4) patient expectations (Figure 1).

### 2.1. Practicalities of Assessing Patients with LRTI

Although the GPs described various organisations of their clinics and different strategies for managing patients with symptoms of acute LRTI, they all described having developed routines for managing this type of patient.

The GPs narrated that the first assessment of these patients was often done over the telephone—either by a staff member or a GP. Some GPs described how time pressure had made them outsource the preliminary examinations to nurses before they assessed the patients themselves and decided on relevant treatment.


*“They (the patients) would typically start with the nurse. She would then take the patients’ history and assess whether a temperature, a CRP test or an oxygen saturation was needed. She would write it down (…) and then I would go in there to see the patient.”*
(GP3)

Other GPs described how the nurses had been trained to evaluate and treat patients themselves, only contacting the GP when in doubt or when patients were deemed severely ill. Yet, other GPs described how they were reluctant to delegate the assessment of patients with acute infections to practice staff and preferred to see these patients themselves due to the possible seriousness of pneumonia, the complexity in considering differential diagnoses, or because they considered lung auscultation to be too difficult to perform for the nurses.


*“They [the nurses] don’t have the same professional background. They don’t have as many differential diagnoses to juggle with.”*
(GP7)

The GPs mentioned a variety of examinations and tests included in the assessment of patients with symptoms of acute LRTI. While CRP tests were available and used in all clinics, not all the GPs had access to the white blood cell (WBC) point-of-care test. Some GPs described how they were reluctant to invest in the equipment because they felt it was too expensive or unnecessary for making a diagnosis. Other GPs, who had already invested in the WBC device, choose not to use it, as the analysis was too time-consuming. The GPs explained how they had organised the workflow to minimise time spend on performing tests. As a result, most GPs described how the tests were mainly performed by staff members.

The point-of-care tests were used at different time points in the consultation. In some clinics, the CRP test was performed upon arrival, so that the result was available for the GP before they saw the patient. A GP described how having CRP and WBC test results available provided a feeling of certainty—although the tests were sometimes deemed redundant. Other GPs expressed concerns about performing CRP tests too often or the test having too much influence on the GPs’ decisions.


*“You may say that we perform more [tests] than what’s actually necessary. It is possible that you sometimes see a patient where you think: I would never have ordered it [the test], if I had seen the patient first.”*
(GP6)

One GP preferred to perform the CRP test himself—and only after a clinical examination deemed it necessary. 

The GPs described how managing patients with symptoms of an acute LRTI could also include microbiological testing, such as sputum samples and throat swabs. However, microbiological tests were not used routinely because of the difficulties with obtaining representative samples, the time delay, and because these tests were often deemed unnecessary. Referral for a chest X-ray was mentioned as an option by some GPs. Still, several GPs described how they usually felt they could manage these patients safely without imaging.

### 2.2. Assessment of the Patient

All the GPs described that their diagnosis was not based only on a single symptom or finding.


*“You can’t just use a single information. It’s a picture you create, based on the things you are told and the things you examine for.”*
(GP2)

Interestingly, the interpretation of patient history, symptoms and findings, and test results differed between the GPs—and each GP had their own diagnostic strategy. Some GPs had difficulties explaining their choices as their actions and decisions sometimes were instinctive or based on a gut feeling. This was especially evident for the evaluation of the patient’s general condition.


*“I think sometimes it can be a little hard to tell why you do as you do. But it’s probably because you see a lot of patients… so I think some of the actions that we do, and decisions, they’re quite subconscious and based a bit on the feeling you get.”*
(GP2)

#### 2.2.1. Patient History, Symptoms and Findings

All the GPs attached great importance to patients’ history, symptoms and the development of symptoms over time. Some symptoms were considered more important than others. The GPs described that patients often complained of coughing. If the cough was dry and prolonged several GPs considered the possibility of an “atypical pneumonia”, while one GP was concerned about a bacterial infection or a more serious disease when the cough was prolonged. The combination of cough and fever was emphasised by all the GPs as an important sign of a bacterial infection. However, one GP highlighted dyspnoea as the most important sign of pneumonia. Many GPs expected the infection to be of viral origin if patients presented with coryza, ear pain or a sore throat. Information about comorbidity, immunosuppression, recent journeys, profession, and prior knowledge of the patient were all used to assess the severity of the infection as well as the possible aetiology. Patients rarely attending general practice were paid extra attention by some GPs as they experienced that these patients usually only contact the clinic when they are very sick.

The GPs described assessing the severity of the infection based on patients’ general condition, and often patients were deemed to have a bacterial infection, if they appeared sick or generally unwell. Nevertheless, the GPs had difficulties explaining how they assessed this “general condition”.


*“You can see, when they are ill. Really, you can. After all they look… they are tired, and perhaps they have a fever. It’s really difficult to explain what it is. Well, you can tell, when a person is seriously ill somehow.”*
(GP3)

Findings such as paleness, clammy skin, difficulties with breathing, delayed patient response, looking ill or feeble were all mentioned as findings to be taken into account. Some GPs explained that it was easier to evaluate these findings when knowing the patient, while others described how measures of blood pressure, pulse, respiratory rate and oxygen saturation sometimes were used to support the evaluation of the patients’ general condition.

Lung auscultation was considered inevitable when examining patients suspected of having an acute LRTI. Auscultation abnormalities described as indicative of pneumonia were crepitation, dullness, and rales—especially if the findings were unilateral.


*“If I hear unilateral crepitation, I of course weigh it highly in relation to pneumonia—if the symptoms fit. If the sounds are mainly located to the lower lobes and bilaterally—I think of pulmonary edema. Those ronchi, which we hear a lot in COPD patients, these sounds are typically more diffuse and spread over both lungs.”*
(GP 6)

However, it differed among the GPs which abnormalities they found most indicative. Although some GPs described how auscultation abnormalities—due to chronic lung diseases—could complicate the interpretation of the examination, most GPs described having great faith in abnormal lung auscultation, and some even valued their findings as decisive. 

#### 2.2.2. Test Results

Point-of-care CRP testing was used by all the GPs as guidance for whether an infection was caused by viruses or bacteria. The GPs described how a low CRP value would confirm a suspicion of a self-limiting infection, whereas a high CRP value could convince them of bacterial aetiology. However, no GPs described using a specific cut-off value for the distinction between a viral and bacterial infection. Some GPs described using CRP tests to monitor changes in the disease course. Few GPs also used a high count of neutrophil granulocytes as a sign of bacterial infection or as a supplement to their interpretation of the CRP test, while several GPs described using oxygen saturation to assess the severity of dyspnoea or to diagnose pulmonary embolism.

#### 2.2.3. Strategies When in Doubt

Diagnostic uncertainty was mentioned by all the GPs—some even said they experienced it rather often, when assessing patients with symptoms of an acute LRTI.


*“The diagnosis is always difficult. I never make a 100 per cent clear-cut diagnosis, I think.”*
(GP7)

When in doubt various strategies were used; additional tests like WBC count or a throat swab were performed, or patients were referred for a chest X-ray. Additionally, most GPs used the chest X-ray examination for excluding other diagnoses, e.g., lung cancer, rather than for verification of pneumonia. All the GPs mentioned the use of a wait-and-see approach, providing a safety net by explaining to patients that they should contact the GP again in case of symptom deterioration or lack of improvement. Some GPs scheduled a follow-up visit a few days later. Still, many GPs also mentioned that sometimes they prescribed antibiotics when in doubt.

### 2.3. Treatment Decisions

No single symptom or finding was identified as decisive for providing an antibiotic course. Importantly, the treatment decision was influenced by several factors, such as assumed aetiology (viral/bacterial), deemed seriousness of the infection, practical circumstances (e.g., home visits), prior knowledge of the patient, test results, patient expectations, and fear of overlooking a serious infection. The GPs sometimes described a pragmatic approach when deciding whether to initiate antibiotic treatment—focusing on the potential need for treatment rather than on providing a specific diagnosis.


*“What makes the exact distinction between a febrile bronchitis and a pneumonia? This is often not clear-cut to me. Rather not, I will risk that a patient gets really sick and ends up in a respirator. Those patients with many risk factors I treat [with antibiotics] immediately.”*
(GP4)

The CRP test was an important piece of the puzzle when deciding whether to treat or not. Some GPs expressed concerns about omitting antibiotic treatment, when CRP levels were high, and a CRP value above 100 mg/L strongly influenced GPs to prescribe antibiotics.


*“Sometimes I see patients where—at first—I don’t think I’m going to prescribe antibiotics. However, if CRP is about 100, I probably shouldn’t consider it too much [prescribing antibiotics]. Then, I shouldn’t have performed the test at all.”*
(GP4)

Some GPs described that if symptoms had evolved rapidly or if patients attended the clinic on a certain weekday (Friday), they were more prone to prescribe antibiotics at a lower CRP level. Fear of consequences, such as a serious course of the disease or even death, if not treating patients with antibiotics were also mentioned by some GPs. 


*“I’d rather provide five or ten unnecessary treatments instead of having one patient dying from it [acute LRTI].”*
(GP1)

Several GPs stated that they were more cautious regarding patients with comorbidities or those having experienced previous severe infections. Sometimes these patients were treated more rapidly and more often with antibiotics than other patients. Importantly, the GPs emphasised that patients with chronic obstructive pulmonary disease (COPD) had to be assessed and treated differently than non-COPD patients.

### 2.4. Patient Expectations

All seven GPs described having experiences with patients expecting an antibiotic prescription when presenting with symptoms of an acute LRTI. Various reasons for expecting antibiotics were mentioned; prolonged duration of symptoms, compromised sleep, previous positive experience with being treated with antibiotics for similar symptoms, and upcoming important events such as a vacation or scheduled surgery. Some GPs mentioned how they sometimes felt pressured or how some patients would not settle before antibiotics were prescribed. 


*“You may state, based on thorough considerations, that the facts are like this.*

*Still, it can be impossible to argue with them [the patients]. And then—sometimes—it’s easier to just give them something [antibiotics].”*
(GP1)

Still, the GPs described how patients most often understood their decisions, when they explained their considerations, the normal course of the disease, risk of adverse events, problems with resistant bacteria, and antibiotics’ lack of effect for treatment of viral infections. Some GPs even described that it had become easier over the years to explain to patients that antibiotics are not always needed. 

Often CRP tests were used as a communicative aid to reassure patients of the viral origin of the infection or to prove the effect of a commenced treatment. Likewise, some GPs described how they used pulse oximetry to convince dyspnoeic patients that their breathing was sufficient, or changes in symptoms to illustrate signs of recovery.

## 3. Discussion

### 3.1. Summary of Main Findings

The interpretation of patient history, symptoms and findings, and test results differed between the interviewed GPs—and each GP had their own diagnostic strategy and routines when managing patients with symptoms of an acute LRTI. Despite having developed routines, the GPs were pragmatic in their approach and sometimes strayed from their usual routine or interpretations as each patient presented different challenges demanding an individual approach. The GPs based their decisions on a general assessment of the patients—both in relation to the assessment of patients, the treatment decisions, and with regard to handling patient expectations. Some factors, such as CRP testing or the patient’s general condition, was included in the assessment across these dimensions. For example, the CRP test was used both to decide on aetiology, seriousness of symptoms and on relevant treatment, as well as to monitor the disease course and as a communicative aid.

All seven GPs described how the diagnosis was not based only on a single symptom or finding, and some GPs had difficulties explaining their choices as their actions and decisions sometimes were instinctive or based on a gut feeling. Moreover, the GPs sometimes described a pragmatic approach when deciding whether to prescribe antibiotics or not—focusing on the need for treatment rather than on providing a specific diagnosis. Importantly, all the GPs mentioned that they had experienced patients expecting antibiotics. Still, the GPs described how patients most often understood the GPs’ decisions, when they explained the pros and cons of antibiotic treatment.

### 3.2. Strengths and Limitations

The qualitative origin of the study enabled an understanding of the opinions, feelings and personal experiences of the participating GPs. As all authors, being medical doctors and some even GPs, were close to the studied field, there was a risk of preconceptions affecting the study results. Thus, to ensure confirmability the following was carried out: coding and analysis were supervised by an experienced qualitative researcher and transcripts were carefully re-read looking for contradictions between transcripts and the analysis. Furthermore, comparison of the analysis with the stated preconceptions was carried out to ensure that new knowledge had emerged.

Only seven GPs were interviewed for this study. Possibly, inclusion of more GPs could have revealed further variations in the GPs’ management of patients presenting with acute cough and provided more detailed information about the GP considerations. However, sampling in qualitative studies are usually performed stepwise and additional participants are included based on knowledge about which information is needed to answer the research question properly [16]. Hence, we recruited the GPs in two steps. As no new analytical themes emerged during the second round of analysis recruitment was stopped. If the last round of analysis had revealed topics or themes that needed further investigation or exploration, more participants would have been recruited.

A say–do discrepancy cannot be ignored when conducting interviews, challenging the credibility of the study. However, the interviewer tried to make the GPs feel comfortable during the interview to enhance credibility. The interviewed GPs appeared relaxed in their normal surroundings as well as honest during the interviews, reporting bad experiences as well as good ones.

Lastly, we did not interview nurses. In clinics where nurses mostly attended patients with symptoms of acute LRTIs, the say–do discrepancy might have been larger as only GPs were interviewed. However, Danish nurses work under delegation of the GP and have to follow the GPs instruction. The GPs, who had outsourced this type of consultation to nurses, still attended these patients in cases of doubt, at home visits and in out-of-hours services. Thus the diagnostic aspect of the management was not missed when solely interviewing GPs.

### 3.3. Discussion of Findings

In this study, we found that most GPs mentioned having practice nurses involved in the management of LRTI patients. The GP’s involvement varied from being involved in every single consultation to solely approving an antibiotic prescription. A recent Danish study found that the GP was involved in about 40% of nurse consultations for acute respiratory tract infections [17]. However, the extent of outsourcing consultations for acute infections to practice staff in Danish general practice is still not fully explored.

The GPs in this study described how different symptoms or findings such as type of cough, dyspnoea or presence of fever, or certain findings, such as lung auscultation abnormalities, were important for making the diagnosis. Back in 1988, Melbye et al. found that a short duration of symptoms before the patient attended general practice (<1 day) was a positive predictor of pneumonia [18]. In a recently published systematic review and meta-analysis by Htun et al. both cough, pyrexia (>38.0 °C), tachycardia (>100/min), tachypnea (≥20/min), and crackles on auscultation were found to be single predictors for a chest X-ray verified pneumonia [19]. Still, guidelines emphasise that no single symptom or clinical finding can be used alone to identify those patients who will benefit the most from antibiotic treatment [5]. Importantly, the general assessment of a patient is deemed as the most important factor, which is also in line with the findings of this present study.

Several studies have shown that the use of CRP testing in patients with acute respiratory tract infections has the potential to reduce antibiotic prescribing significantly [20]. In this qualitative study it was described how point-of-care tests, were sometimes performed even before the patient was clinically assessed by a nurse or a GP. This approach is not fully in line with existing national recommendations [5], in which a CRP test is recommended to be used only when diagnostic uncertainty exists after history taking and a clinical examination. However, not only Danish GPs seem to use the CRP test frequently. A Swedish study has also found an increased use of CRP testing for patients with symptoms of an acute LRTI from 55% in 2006 to 62% in 2014 [21].

Most of the GPs did not refer patients for a chest X-ray when they presented with symptoms of an acute LRTI. This finding is in accordance with a previous Danish study, in which only 6% of patients diagnosed with pneumonia had a chest X-ray performed [14]. Additionally, it has been shown that Danish GPs refer for a chest X-ray less frequently than Spanish GPs before deciding on the diagnosis pneumonia [15]. It is possible that diagnostic accuracy of pneumonia could be increased if GPs used imaging more frequently. Nevertheless, a recent Swedish study found that making a clinical judgment regarding patients with suspected pneumonia is strongly associated with a chest X-ray confirmed infiltrate. Consequently, if the GP is sure of the diagnosis there seems to be no apparent need to refer patients for a chest X-ray [22].

The GPs described a pragmatic approach, when managing patients with symptoms of an acute LRTI. An overall focus on “the need for antibiotic treatment or not” rather than on labelling the patient with a specific diagnosis was present through the interviews. Much evidence concerns healthcare professionals’ ability to differentiate diagnostic features of pneumonia from those of acute bronchitis [14,18,23]. In 1972, Howie stated that GPs did not always follow the ordinary path going from symptoms—to a diagnosis—to a decision on treatment. Instead he argued that GPs jumped directly from symptoms to the treatment decision, interposing the diagnosis later as a justification for a treatment [24].

Importantly, the weighing of benefits and harms always needs to be taken into account when managing patients with acute RTIs. However, a study by Gulliford et al. identified that even a quite large reduction in antibiotic use in general practice for self-limiting RTIs was only associated with a slight increase in the incidence of pneumonia and peritonsillar abscess [7]. Additionally, the number of other complications such as mastoiditis, meningitis and empyema was not increased despite antibiotic use was lowered [7]. No single symptom or finding can fully establish the diagnosis, nor the prognosis, of acute LRTIs and differences exist in GPs management of this group of patients. This diagnostic uncertainty often results in an overuse of antibiotics for patients presenting with acute cough. Future studies must focus on developing and validating clinical prediction rules to identify those patients, who will benefit the most from antibiotic treatment. Better prediction rules and guides to interpretations of tests and examinations could also help standardise the outsourcing of assignments to practice staff, making them less dependent on their gut feeling. As shown in this study, the use of a CRP test may serve several purposes and the use has increased enormously in Danish general practice over the past decades. Clinical practice could benefit from more studies focusing on when the CRP test is appropriate to use and when clinicians are better off without the result of the test.

## 4. Materials and Methods

An explorative phenomenological approach [25] was used in semi-structured interviews to gain access to the GPs’ lifeworld and their lived experiences with handling patients with symptoms of acute LRTI. All interviews were conducted and transcribed by the first author Lise Bisgaard (LB), who was a female medical student, with no prior experience in performing qualitative studies. Before conducting the interviews LB’s presuppositions were stated in a document. Thus, an awareness of these conceptions was present during the interviews and the analytical process and eventually enhanced the ability to assess the credibility and reflexivity of the following results. The qualitative checklist, Standards for Reporting Qualitative Research (SRQR), was followed [26].

### 4.1. Setting

Denmark has a public healthcare system, where GPs (medical doctors, with a minimum of six years of specialisation in family medicine) provide primary healthcare for patients on their list. Most practices also employ nurses, and in some practices, they also attend patients with acute infections. In Denmark, nurses can prepare an antibiotic prescription in the electronic system. However, GPs are obliged to approve the scripts before the patient can redeem them at a pharmacy. Danish citizens have free access to general practice when they are listed with a GP, which more than 99% are [27]. GPs are paid through a combination of a fixed capitation fee per listed patient and fee-for-services, including point-of-care testing like CRP and WBC count [28,29]. If imaging is deemed indicated the patient will be referred to a nearby hospital or an X-ray clinic. 

### 4.2. Recruitment

Prior to the study, we estimated that between five and eight GPs were needed for this qualitative study—taking the concept of information power into account [30]. To achieve variation, GPs from both urban and rural locations in the North Denmark Region, with varying age, experience, gender, and practice organisation were approached. We assumed that GPs in practices with many nurses employed would have different experiences than GPs in practices with only a few—or even none. Hence, the number of nurses working in the practice was taken into account.

Participants were recruited stepwise. The GPs were invited by a personal letter, and subsequently contacted by phone by LB asking for participation. First, six GPs were purposely selected based on background characteristics found on the Danish eHealth Portal [31] and at the clinics’ homepages. Four GPs agreed to participate (GP 1, 2, 3 and 4). After initial analyses of these four interviews, the research team found that the obtained interviews included a high amount of information. Another four GPs were then purposely selected based on reflections from the first round and information from the Danish eHealth Portal and at the clinics’ homepages. Three of these GPs agreed to participate (GP 5, 6 and 7). As no new analytical themes emerged in the analysis of these interviews, recruitment was stopped. In total 10 GPs were invited for the study. Seven GPs were interviewed, while three GPs declined the invitation due to time pressure.

Before each interview the overall aim of the study was briefly explained to the participant and information about the research team (names, titles) was provided. Participation was voluntary, and each GP received an economic compensation corresponding to one hour of their time [32]. The interviews were conducted from January to March 2020. Importantly, all interviews were performed before the COVID-19 pandemic reached Denmark.

### 4.3. Interviews

An interview guide was developed from available literature and informal discussions within the research team. The overall themes for the guide were: (1) Management of a patient presenting with acute cough, (2) when to suspect pneumonia, (3) antibiotic treatment, (4) diagnostic doubt, and (5) influence of physical surroundings and time pressure on decision making. The interview guide provided a structure and suggested questions whilst still allowing the interviewer to follow and explore the GPs’ narratives. The interview guide did not evolve over the course of the study.

To make the GPs feel at ease in their natural working environment and to help them maintain their role as healthcare professionals, the interviews were conducted in the GPs’ offices. The GP and the interviewer were the only persons present during the interview. Individual interviews gave the GPs the opportunity to share their experiences and account for their behaviour and decisions without having concerns about their colleagues’ opinions, judgment or critique.

### 4.4. Data Analysis

All interviews were audio-recorded and transcribed verbatim by LB. No field notes were made, and transcripts were not returned to the participants for comments. The transcribed interviews were analysed using systematic text condensation [33], which is a thematic cross-case analysis conducted in four steps: (1) total impression, (2) identifying and sorting meaning units, (3) condensation, and (4) synthesising. LB conducted the analysis supervised by other members of the research group, who were all medical doctors (two GPs, two full-time researchers). The analytic process is demonstrated in Figure 2. Interviews and analyses were conducted in Danish and translated into English after the final analysis.

## Figures and Tables

**Figure 1 antibiotics-10-00661-f001:**
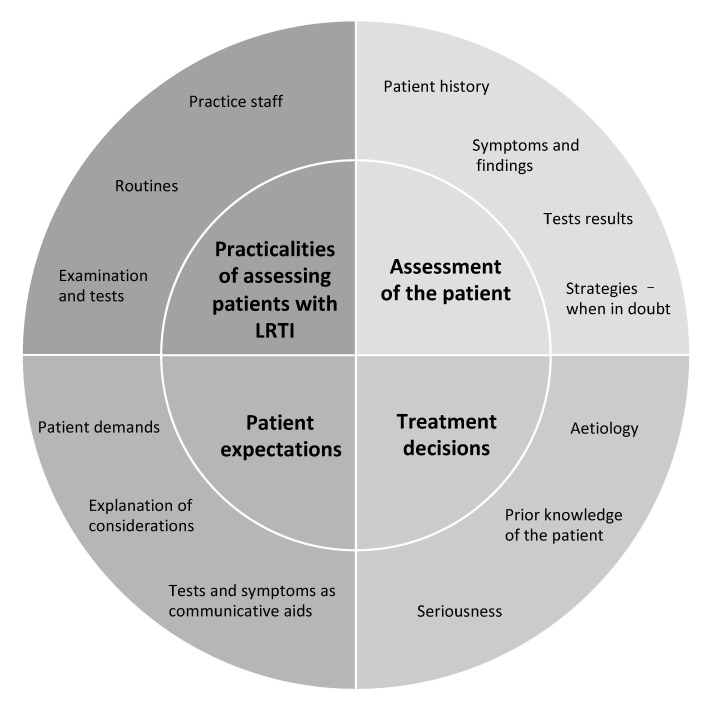
Overview of the analytic themes and the components within each theme.

**Figure 2 antibiotics-10-00661-f002:**
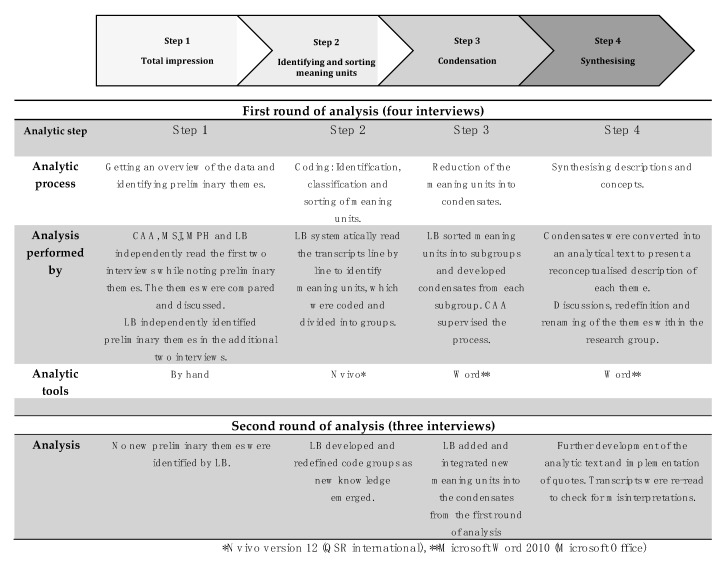
Overview of systematic text condensation in this study.

**Table 1 antibiotics-10-00661-t001:** Characteristics of GPs.

GP	Gender	Age, Years	Experience As a GP, Years	Clinic Location	Type of Clinic	Nurse Involvement *
1	M	≥65	>20	Rural	Solo	II
2	M	45–54	11–15	Urban	Partnership	I
3	F	≥65	16–20	Rural	Partnership	II
4	F	55–64	16–20	Urban	Solo	I
5	F	<45	<5	Rural	Solo	III
6	F	<45	5–10	Rural	Partnership	III
7	M	45–54	5–10	Urban	Partnership	I

M = male; F = female. * I = GP attend patients without nurse involvement, II = Nurse attend patients before GP assessment, III = Nurse mostly attend patients without GP involvement.

## Data Availability

The data presented in this study are not publicly available due to privacy restrictions.

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
