# Peer review of "Danish GPs’ Experiences When Managing Patients Presenting to General Practice with Symptoms of Acute Lower Respiratory Tract Infections: A Qualitative Study"

_antibiotics, 2021, doi:10.3390/antibiotics10060661_

Round 1
Reviewer 1 Report
This is a well written and interesting paper addressing the important important issue of antimicrobial stewardship in primary care in relation to the treatment of LRTI.
The methods are appropriate and well described and results clearly presented.
The findings are a useful base upon which to inform the design of effective intervention to support GPs in decision making around antibiotic prescribing.
I would suggest that in the introduction reference is made to important work by Gulliford et al. BMJ 2016;354:i3410 looking at the safety of reduced antibiotic use for self limiting RTI.
Minor rewording
suggest line 365 - say 'making a clinical judgement regarding patients......'
suggest 429 say 'the overall aim of the study was briefly explained to......'
The final conclusions are valid - I would suggest again a little more discussion of the safety/risks and benefits of reduced antibiotic use in this context
Reviewer 2 Report
The manuscript focuses on an important and timely health care issue and I enjoyed reading it. In general, the study is well described. There are several major and minor revisions that would enhance the study manuscript.
Introduction:
- The introduction is nicely structured and well written.
Results:
- The results are clearly structured. I have two comments:
- It is always written by “all doctors”. In the method later it turns out that there are 7 doctors. With "all doctors" everyone always thinks of more than 7. Please also write in the results that all 7 doctors are meant.
- 2.1 “Although the GPs described various organisations of their clinics…” - Does this actually mean clinic or practice?
Discussion:
- Line 289 ff: “The interpretation of patient history, symptoms and findings, and test results differed between the interviewed GPs – and each GP had their own diagnostic strategy and routines when managing patients with symptoms of an acute LRTI.” The described difference is nowhere made clear, to what extent did the doctors differ from each other? The analysis rather emphasised the similarities.
- 3.2 Strengths and Limitations: Here I miss the discussion of the very small number of doctors interviewed. Although this is addressed in the methods, is it not conceivable that with further interviews other patterns will emerge - especially in view of the emphasised differences of the doctors studied (Line 151 f). I think with 7 doctors this cannot be ruled out.
Methods:
- Line 411 f: “GPs from the North Denmark Region were invited. All GPs were invited by a personal letter, and subsequently contacted by phone.” How many doctors were contacted exactly? The seven interviews appear to be few - even for a qualitative study. But if only 20 doctors are active in the region and were contacted, then that would be plausible.
- Line 419-428: The procedure described here seems to contradict that from line 411-412. However the recruitment is not clear and the part would have to be revised again.
- Line 436-437: “The interview guide was pilot tested by a GP, who was also a member of the research group.” If the GP was part of the development group, surely the piloting cannot be done with him. In my view, this has to be changed.
- 4.3: What was the structure of the interviews like? Which topics were named?
- When presenting the method, it makes sense to use existing checklists, e.g. COREQ (https://www.equator-network.org/reporting-guidelines/coreq/).
Round 2
Reviewer 2 Report
Many thanks for incorporating the suggestions.